Resource

# A practical comparison of the next-generation sequencing platform and assemblers using yeast genome

Min-Seung Jeon[1], Da Min Jeong[1], Huijeong Doh[1], Hyun Ah Kang[1], Hyungtaek Jung[2], Seong-il Eyun[1]

Assembling fragmented whole-genomic information from the sequencing data is an inevitable process for further genome-wide research. However, it is intricate to select the appropriate assembly pipeline for unknown species because of the species-specific genomic properties. Therefore, our study focused on relatively more static proclivities of sequencing platforms and assembly algorithms than the fickle genome sequences. A total of 212 draft and polished de novo assemblies were constructed under the different sequencing platforms and assembly algorithms with the repetitive yeast genome. Our comprehensive data indicated that sequencing reads from Oxford Nanopore with R7.3 flow cells generated more continuous assemblies than those derived from the PacBio Sequel, although the homopolymer-based assembly errors and chimeric contigs exist. In addition, the comparison between two second-generation sequencing platforms showed that Illumina NovaSeq 6000 provides more accurate and continuous assembly in the second-generation-sequencing–first pipeline, but MGI DNBSEQ-T7 provides a cheap and accurate read in the polishing process. Furthermore, our insight into the relationship among the computational time, read length, and coverage depth provided clues to the optimal pipelines of yeast assembly.

## Introduction

Genome projects employ state-of-the-art DNA sequencing, mapping, and computational technologies to understand molecular/cellular mechanisms, gene repertoires, genome architecture, and evolution (Jung et al, 2020b). As most of these analyses require genomic structures or continuous genomic sequences at the chromosomal level, upgraded sequencing technologies and assembly tools have made whole-genome reconstructions more accurate and efficient. However, the unprecedented output of short- or long-read sequencing technologies rendered the consecutive assembly process out of date, suggesting the necessity to maximize advantages and minimize shortcomings of sequencing reads (Di Genova et al, 2021).

Thus, the ideal de novo genome assembly is fully required to understand the characteristics of sequencing technologies and the operating principles of the assemblers.

In terms of sequencing platforms, current genomic research has accomplished remarkable growth with the advent of second-generation sequencing (SGS) using massive parallelization of amplification, each based on its own clonal DNA template (Goodwin et al, 2016; Giordano et al, 2017). With this property, SGS technology features high-throughput and fast, cheap, and highly accurate reads. Currently, the most widely used short-read platforms are based on sequencing technology by Illumina and by the Beijing Genomics Institute group, which have developed the up-to-date Illumina NovaSeq 6000 and the MGI DNBSEQ-T7 sequencing platform, respectively (Goodwin et al, 2016; Logsdon et al, 2020; Kim et al, 2021). Although showing up to 99.5% overall read accuracy, the Illumina sequencing platform may produce substitution errors. The additional problem of the under-representation of high or low guanine–cytosine (GC) regions has also been reported, even in highly repetitive genes; this subsequently generates gaps in assembled contigs, requiring a read depth of more than 50× without a GC bias (Chen et al, 2013; Goodwin et al, 2016; Bainomugisa et al, 2018; Browne et al, 2020; Jung et al, 2020b). Furthermore, short-read–only assemblers cannot resolve highly heterozygous sequences in repetitive regions, presenting difficulties in distinguishing the genome-wide structural variants and haplotypes (Phillippy et al, 2008; Chaisson et al, 2015; Liem et al, 2017; Jung et al, 2020b; Tedersoo et al, 2021). Thus, those properties can provoke the cessation of contig extension at the border of repeat regions even under high-coverage conditions, resulting in misassembly.

Fortunately, third-generation sequencing (TGS) has resolved these problems by providing a read length of about 10–20 kbp (up to even thousands of kbp), which is often longer than the genomic repeat areas (Rhoads & Au 2015; Giordano et al, 2017; Kolmogorov et al, 2019; Kumar et al, 2019; Jung et al, 2020a; Giani et al, 2020). The single-molecule real-time (SMRT) sequencing method commercialized by Pacific Biosciences (PacBio) directly reads native DNA sequences rather than clonal amplification, which makes SMRT sequencing less sensitive to GC contents than sequencing on other platforms (Reuter et al, 2015; Goodwin et al, 2016; Hebert et al, 2018). Also, generating a circular template that can be sequenced by polymerase in multiple cycles improved the sequencing quality of SMRT

---

[1]Department of Life Science, Chung-Ang University, Seoul, Korea   [2]Queensland Alliance for Agriculture and Food Innovation, The University of Queensland, St Lucia, Australia

Correspondence: hyungtaek.jung@uq.edu.au; eyun@cau.ac.kr

technology (Travers et al, 2010; Reuter et al, 2015). Another market-leading long-read sequencing platform provided by Oxford Nanopore Technologies (ONT) detects specific electrical disruption when denatured single-strand DNA passes through the nanopore on a flow cell. This technology provides a varying range of throughput levels according to the number of flow cells (MinION, GridION, and PromethION) (Urban et al, 2015 Preprint; Goodwin et al, 2016; Logsdon et al, 2020). Because the error rate of a 1D read (forward or reverse read) approaches about 30% of mainly indel errors, ONT MinION increases accuracy using a double-checking system of DNA strands connected by hairpin adapters that can pass through the pore in regular order, thereby resulting in 2D reads (Goodwin et al, 2016). However, compared with the SGS platforms, the TGS platforms still have raised concerns about their higher sequencing error rates (5–20% for TGS and 1% for SGS), which can negatively affect assembly accuracy (Goodwin et al, 2016; Giordano et al, 2017).

In addition to the error proclivity of these sequencing platforms, the assembly algorithms are also an influential factor. The complete whole-genome assembly can be acquired by verifying all of the alignment blocks and overlapping probability of input sequencing reads. However, this methodology also dramatically increases the complexity of the process. Therefore, different assembly tools compromise their output quality and efficiency using different algorithms for their main purpose. In other words, the well-fitted sequencing platforms and assembly algorithms create synergy in the assembly process through a combination of sequencing technology and specific programs. The interaction between the sequencing platform and assemblers has critical effects on outputs and can, if not carefully matched, lead to deteriorations in assembly quality.

The TGS-only assemblers represented by Flye, WTDBG2, and Canu differ in their assembly process that is adapted to their main purpose. Flye constructs disjointigs from raw TGS reads, and its graph-based repeat resolution of bridged and unbridged repeats leads to a contiguous assembly process that requires little computational time (Kolmogorov et al, 2019). WTDBG2, which focuses on the fastest possible assembly, drastically reduces the computational cost by constructing a hash table with homopolymer-compressed read units and extends contigs based on the analogy of the de Bruijn graph (DBG) using a compressed sequence (Ruan & Li, 2020). In contrast to the aforementioned direct assemblers, Canu assembler's main purpose is to ensure a highly accurate assembly and thus to incorporate multiple rounds of error correction (Koren et al, 2017). Similar to WTDBG2, this assembler extends contigs with a hash table based on $k$-mer spectra, but an overlap comparison between TGS reads in a repeat-sensitive manner permits contig construction in a greedy fashion.

However, assembly derived from relatively high error rates such as TGS platforms can confuse the exact location of the base in the whole genome, thereby leading to insertion and deletion errors (indels) (Zimin et al, 2017; De Maio et al, 2019; Jaworski et al, 2020). The MaSuRCA algorithm, therefore, extends accurate SGS reads to their maximum unique length and connects these "super-reads" based on partial alignment with the long erroneous TGS reads in a greedy fashion (Zimin et al, 2013, 2017). Another recently developed hybrid assembler, WENGAN, similarly pre-assembles SGS reads with DBG and deduces the location of each contig with the pseudo-formed paired-end TGS alignment (Di Genova et al, 2021).

Our analysis also compared the performance of SGS platforms (Illumina and MGI), which are cheaper than TGS platforms, as checking whether that read can be a sole sequence information provider in the de novo whole-genome assembly. The SPAdes, which can proceed through hybrid and non-hybrid (short-read–only) assembly pipelines, measures its quality metrics for comparison with the TGS assemblers. The performance of the multisized DBG, which contributes to improving contiguity compared with the conventional DBG, was verified in a previous study (Bankevich et al, 2012). ABySS uses the conventional DBG and extends their unambiguous contigs to the scaffold, looking up the paired-end information of sequencing dataset (Simpson et al, 2009).

Here, we sequenced and assembled the whole genome of a yeast species, *Debaryomyces hansenii* KCTC27743, using four different sequencing platforms (PacBio Sequel, ONT MinION, Illumina NovaSeq 6000, and MGI DNBSEQ-T7), to assess pipelines from pre-assembly to the reconstructed whole-genome assembly analysis. The model yeast *D. hansenii* has both haploid and diploid genomes and is a cryotolerant, osmotolerant, and xerotolerant biotechnological yeast (Jeong et al, 2022). It is prevalent in the salty aquatic environment and food products and thus used as a flavoring agent for producing volatile aldehydes and alcohols (Sørensen et al, 2011; Flores et al, 2017). It is also a promising probiotic yeast whose β-glucan stimulates immune responses in humans, fish, and mice (Angulo et al, 2020; Jeong et al, 2022). The whole-genome assembly of *D. hansenii* was conducted first by Dujon et al (2004), demonstrating enriched gene duplications in *D. hansenii* compared with *Saccharomyces cerevisiae*, *Yarrowia lipolytica*, *Kluyveromyces lactis*, and *Candida glabrata* (Dujon et al, 2004). The heterogeneous genome structure of *D. hansenii* strains was analyzed in several studies (Petersen & Jespersen, 2004; Jacques et al, 2010). However, despite the relatively small size of the yeast genome compared with other eukaryotic genomes, the most effective whole-genome sequencing method and assembly tools for yeast genomes have not yet been determined. In this study, our goal was thus to propose standards for the optimal sequencing platform, including minimal coverage depth and a well-fitting assembly strategy, for the de novo yeast assembly. To that end, a total of 212 de novo assemblies of the yeast *D. hansenii* KCTC27743 were processed using four different sequencing platforms, six different read depth levels, and seven state-of-the-art assembly tools with different algorithms. Our analysis of assembly results could widen the prospects of such algorithms and give insights into strategies for complete genomic reconstruction by comparing the differences in results with three different aspects: sequencing platforms, coverage depth, and assembly programs.

## Results

### Genomic property estimation based on $k$-mer spectra

We first generated the sequencing reads of our yeast (*Debaryomyces hansenii* KCTC27743) with two TGS platforms (PacBio Sequel and ONT MinION) and two SGS platforms (Illumina NovaSeq 6000 and MGI DNBSEQ-T7). Fig 1 shows the schematic pipeline of the

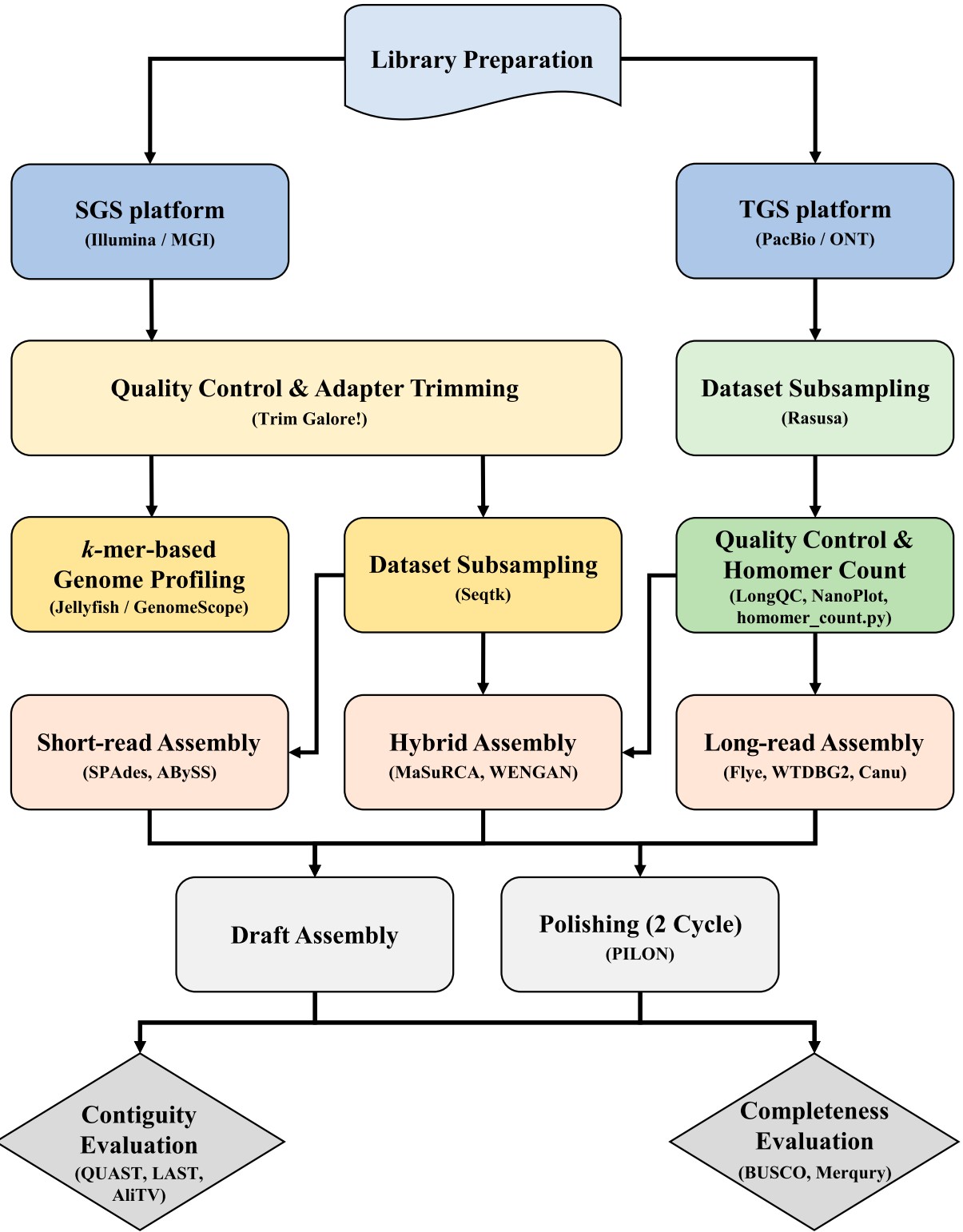

**Figure 1. Flowchart of the sequencing, assembly, and evaluation.**
The depicted pipeline is composed of three main components: (1) sequencing process on different platforms (PacBio Sequel, ONT MinION, Illumina NovaSeq 6000, and MGI DNBSEQ-T7); (2) whole-genome assembly at different coverages (20×, 30×, 40×, 50×, 60×, and 70×); and (3) different assembly programs (Flye, WTDBG2, Canu, MaSuRCA, WENGAN, SPAdes, and ABySS).

overall assembly process using the combination of the sequencing dataset. Before the assembly, we estimated the genome size, which is a required option in most assembly tools, and checked the draft genomic properties such as heterozygosity, repeat rate, and read error rate. As GenomeScope requires unidirectional support for highly accurate short-read platforms (Ranallo-Benavidez et al, 2020), two quality-controlled SGS datasets (2.08 and 1.29 Gbp for the Illumina and MGI, respectively) were used with GenomeScope (Table S1). The constructed $k$-mer profiles provided information about presumptive genome size and level of heterozygosity (Fig S1A and B). Among the total 10 $k$-mer models ($k$ = 15, 17, 19, 21, and 23 for Illumina and MGI reads, respectively), the Illumina dataset provided the estimated genome size (12.53 Mbp) with the best model-fit ratio (99.32%) at $k$ = 15 (Table S2). The shape of the $k$-mer distribution also suggests the haploidy of *D. hansenii* KCTC27743, considering the higher remarkable peak in the second dotted line that indicates that most reads belong to unique homozygous sequences (Fig S1A) (Vurture et al, 2017). The hollowed-out peaks in the first and third dotted lines are concordant with extremely low heterozygosity (max 0.00488138%) of this species (Table S2). Thus, this $k$-mer spectrum represented *D. hansenii* KCTC27743 as a haploid organism with rare nucleotide divergence. Moreover, considering that most yeast species have repetitive DNA in the range of 1–25%, the level of genome repeat length even with a short 15-mer (12.64%) indicates that our isolate contains a relatively high ratio of repetitive regions (Rao et al, 2018).

## De novo assembly pipelines on different sequencing platforms

Both SGS and TGS datasets of *D. hansenii* KCTC27743 were subsampled to a constant size to minimize sample size–associated bias. As the Illumina and MGI sequencing reads were constant in the read length (150 bp) but not in total read number (7.09 × 10$^6$ and 5.01 × 10$^6$, respectively), 4 million reads were subsampled for both SGS sequences. In contrast to the SGS sequences, the TGS sequences were highly variable in read length and thus subsampled, taking both total read and base number into consideration. As the size of PacBio is about four times larger than that of the ONT platform (16.45 and 4.31 Gbp, respectively), we randomly subset each platform's read into six coverages (20×, 30×, 40×, 50×, 60×, and 70×) for an accurate comparison (Fig S2). The metrics of the sequencing data in the subsets of both TGS platforms revealed consistency in each subset regarding the average read length, N50, and GC content, whereas the read number and total base increased at regular intervals according to the increase in coverage (Table S3). Two hybrid and five non-hybrid assemblers proceeded through the whole-genome reconstruction of 88 draft assemblies, and two cycles of the polishing process by PILON further produced 124 short-read–based error-corrected assemblies from two SGS platforms. Because PILON tends to underestimate tandem repeat resolution in short-read–based correction processes (Walker et al, 2014), some of our polished assemblies also reduced their total length compared with the draft assembly results. To further investigate the completeness of the draft and polished assemblies, we compared both BUSCO and Merqury scores for each assembly and represented the metrics according to the hybrid and non-hybrid methods.

## Completeness analysis of non-hybrid assemblers

The BUSCO completeness values according to the coverage depth in Fig 2A indicate that the erroneous long-read–only draft assemblies obtained with Flye, WTDBG2, and Canu tended to increase in quality when more TGS data were provided as expected. The completeness performance from the three TGS assemblers showed better initial BUSCO completeness for PacBio subsamples than ONT subsamples, scoring a minimal 50.8% and 12.3% on WTDBG2 at 20×, respectively. As previously reported, the homopolymer error-prone property leads to a serious assembly accuracy reduction in the nanopore R7 device, whose signal can be more easily disturbed by more nucleotides passing through pores than that of the latest R9 or R10 device (Liem et al, 2017; Rang et al, 2018; Sereika et al, 2022). We also detected the number of homomers in the TGS datasets and discovered that the number of homomers in the PacBio reads is twice than that of ONT reads (1.6% and 0.8%, respectively) (Table S4). WTDBG2 showed the lowest accuracy on both TGS platforms and particularly lower accuracy in nanopore datasets, because of its lack of error correction and its use of only one consensus step compared with Flye and Canu (Ruan & Li, 2020). These findings comprehensively indicated that the fast assembly process of WTDBG2 was certainly worth it, but that the completeness of WTDBG2 using a sole TGS platform clearly depends on the accuracy of the raw data. In contrast to the draft assembly results, the correction stages of both SGS reads significantly increased sequential fidelity regardless of the TGS datatype. That means that even if the accuracy of the sequencing dataset is much lower, it is sufficiently complemented by secondary sequencing on platforms such as Illumina and MGI. This insight can be supported by the fact that the ONT-based Canu assembly polished by MGI reads was the best in terms of chromosomal structure, contiguity, and completeness among all hybrid and non-hybrid assemblies. However, in contrast to some of these significant differences between assemblers, SPAdes and ABySS did not show shifts in BUSCO completeness (SPAdes: 85.9%/84.2% and ABySS: 90.0%/75.1% on the Illumina/MGI, respectively) after the polishing process, representing a slight increase in genome size (Table S5).

## Completeness analysis of the hybrid assemblers

Fig 2B shows the BUSCO completeness comparison of the hybrid assembly programs according to their coverage. Most of the hybrid assemblies yielded high and constant BUSCO values regardless of SGS-mediated polished and exhibited comparable BUSCO completeness to the PILON-polished non-hybrid assembly results. Although there was a negligible completeness difference in the draft and error-corrected assemblies, MaSuRCA yielded a few drastic decreases in TGS subsamples even when read depth was increased (PacBio + Illumina 60× and PacBio + Beijing Genomics Institute 50×). The further study of synteny analysis represented this sudden drop in BUSCO value was caused by the severe fragmentation of the scaffold in the assembly procedure (Fig S3). In addition, the combination of ONT reads and MaSuRCA assembler generated a higher BUSCO duplicate ratio (0.1–4.4%) than the PacBio assembly (0.0–0.9%) (Fig 2C). Both the MaSuRCA and WENGAN algorithms start with an SGS-based pre-assembly process, but their

(A)

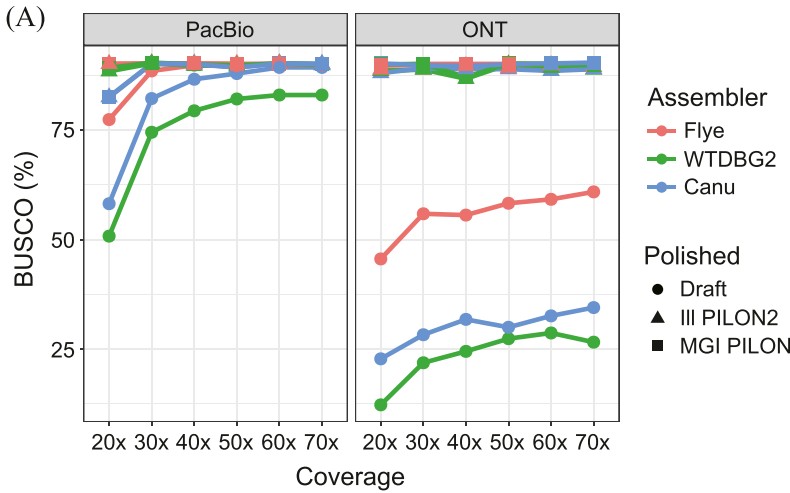

**Figure 2. Comparisons of draft and polished assemblies from hybrid and non-hybrid assemblers.**
**(A)** Comparison of BUSCO completeness among TGS-based non-hybrid assemblers. The assembly results of Flye, WTDBG2, and Canu were compared according to the polishing method and are expressed as ratios of complete genes. **(B)** Comparison of BUSCO completeness between hybrid assemblers. The presented data are draft results that did not substantially differ from polished data. **(C)** Numbers of duplicated BUSCO genes in the seven different assemblers.

(B)

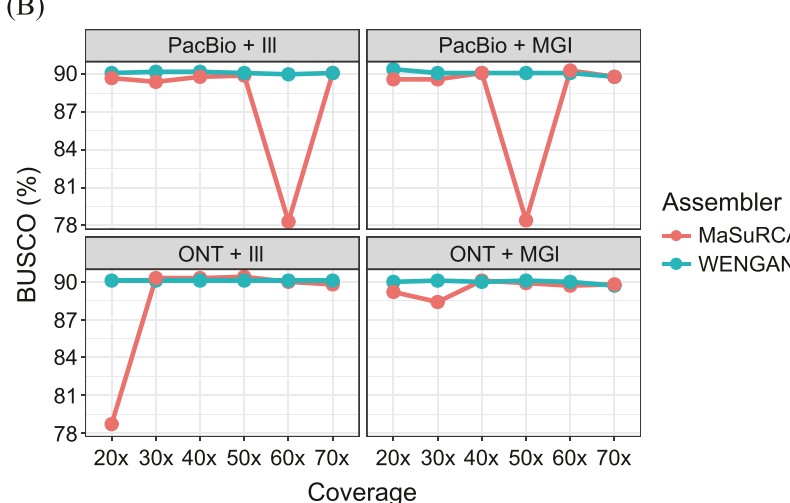

(C)

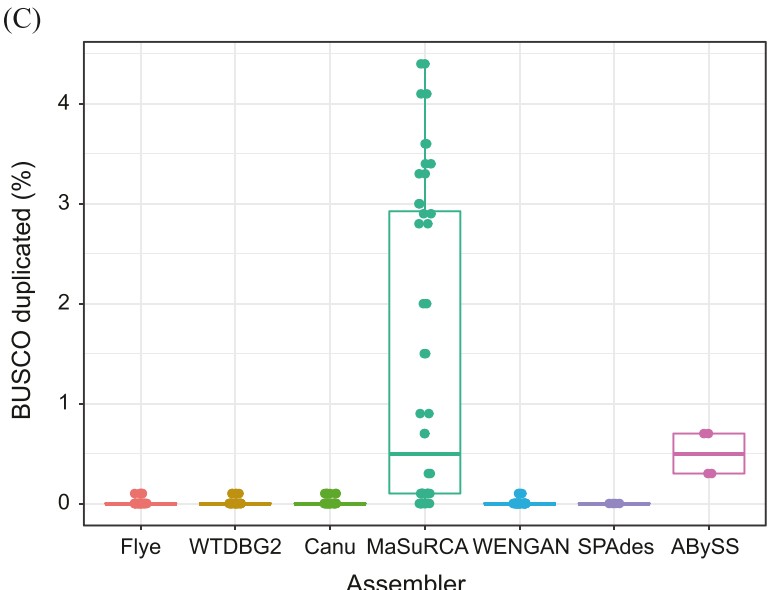

methods for contig extension are divided into branches as mentioned earlier. MaSuRCA greedily extends the contig by referring to the long-read sequence based on the unique read that can be derived from each super-read, but WENGAN minimizes the likelihood of repeating the DBG-assembled short-read contigs in the subprocess of detecting chimeric contigs using pseudo–paired-end alignment of long reads (Zimin et al, 2017; Di Genova et al, 2021). The occurrence of BUSCO duplication shows an increasing tendency towards higher coverage of TGS reads, which is exacerbated as the accuracy of TGS reads drops.

### Chromosomal structure of the assemblies

The strength of traditional metrics for assembly contiguity such as N50 (the length of contigs corresponding to half of the total length when ordered from largest to smallest) and contig number lies in their capacity to quickly capture the quality of the genomic structural resurrection. In this respect, short-read–only assemblies from the SPAdes and ABySS revealed a hard-to-resolve property issue with repeat elements, which are sufficiently longer than the read length. As a result, SGS-based whole-genome assemblies formed more fragmented contigs than those of TGS-based: 31 and 37 contigs in the Illumina-based assembly and 37 and 44 contigs in the MGI-based assembly in SPAdes and ABySS assemblies, respectively. However, we also found some false merged contigs in TGS-based assemblies compared with the reference genome of *D. hansenii* CBS767, a haploid genome (12.18 Mbp) consisting of 7 chromosomes (Fig S4) (Dujon et al, 2004), which was caused by the erroneously connected repeat regions. The chimeric scaffolds seen in Fig 3A and B and the assembly metrics seen in Table 1 depict the end-to-end connection from chromosome to chromosome for the Flye, WTDBG2, MaSuRCA, and WENGAN. This phenomenon seems to stem from a failure of repetitive sequence resolution at the end of the chromosome, because it is hard to distinguish between highly variable yeast telomeres and long continuous repetitive regions. Moreover, the relatively error-prone ONT dataset generated more chimeric contigs, which were merged in contig ends than the PacBio dataset. However, Canu never showed a chromosomal fusion in the telomeric region because of a strict criterion for repetitive sequence connections than other assemblers (Fig 3C). In addition, the SPAdes, another chromosomal fusion-free assembler, exhibited a lack of chimeric contigs similar to the Canu, but more fragmented contigs, especially in the telomeric region indicating the limitations of SGS-only assembly pipelines in long repetitive regions.

### Complexity reduction of the assemblies

Differences in the laboratory environments of computational genomic assemblies point to the issue of adequate experimental time cost. Complexity reduction algorithms embedded in assembly programs allow users to reduce the spatial (memory) and temporal (computational time) scale of calculations. We therefore recorded the real wall time for all our assemblies and evaluated the feasibility of assembly pipelines as pre-processors of further genomic analyses in Fig 4. The three TGS-only assemblers (Flye, WTDBG2, and Canu) exhibited a larger difference in computational time between assemblies with the PacBio Sequel and the ONT MinION than the other four assemblers (MaSuRCA, WENGAN, SPAdes, and ABySS). We furtherly examined the time consumption of assembly stages, the correction stage and the

contig extension stage, in non-hybrid assemblers (Fig S5 and Table S6). The time consumption differences in the contig extension stage were more pronounced than those in the correction stage on the two platforms (WTDBG2 and Flye) using the same coverage dataset. Note that WTDBG2 does not proceed with the error correction stage and Flye proceeds with polishing at the end. In addition, the time difference in the correction stage using the same coverage dataset was shorter than that in the contig extension stage in Canu having repeatedly performed corrections. In non-hybrid assembly, therefore, the factor influencing the contig extension stage affects the temporal difference between the two platforms more than the factor affecting the correction stage. WTDBG2 showed particularly low time consumption and was least affected by the differences in read length between the two TGS platforms because it compressed the erroneous TGS reads using not only homopolymer compression but also dynamic pairwise alignment programming. These processes simplify the 256-bp length of each sequence into one binned unit and thereby dramatically decrease time complexity compared with the base- or $k$-mer–level calculation (Smith & Waterman, 1981; Berlin et al, 2015; Ruan & Li, 2020). In the case of both hybrid assemblers (MaSuRCA and WENGAN), which start their assembly pre-processes with a short SGS read, there was little difference in the time consumption of the assembly process between the platforms, indicating that the contig extension stages using long TGS reads in their assembly process were hardly affected by the amount or length of the TGS reads.

## Discussion

We employed long- and short-read–based assemblies under various conditions and compared them in terms of sequencing platforms, coverage depth, and assembly programs. Until today, a plethora of studies about dynamic genome reconstruction has been published that compare non-unified assembly processes (Fournier et al, 2017; Giordano et al, 2017; Murigneux et al, 2020). As Magoc et al (2013) suggested, however, the optimal assembly pipeline can vary according to the properties of biological sequences, even in low-complex bacterial genomes, and it can therefore be difficult to adjust the system to the proper genomic reconstruction, depending on which species is analyzed (Magoc et al, 2013). We thus focused on the inherent properties of the sequencing and assembly methods rather than the properties of the organism. This perspective provided the observation of the consistent assembly results according to three different categories of the assembly algorithm: (1) completeness (error correction) algorithm, (2) contiguity (contig extension) algorithm, and (3) computational efficiency (complexity reduction) algorithm (Table 2).

In terms of the TGS platforms, assemblies based on highly accurate PacBio reads from non-hybrid assemblers yielded higher BUSCO completeness as expected. Despite their sole usage of error-prone TGS reads, Flye and Canu reached completeness levels similar to those of their polished results at particular coverage thresholds (40× and 60×, respectively). Those draft assemblies, which were close to completion, suggest that both assemblers have the capacity for following genomic analyses such as gene

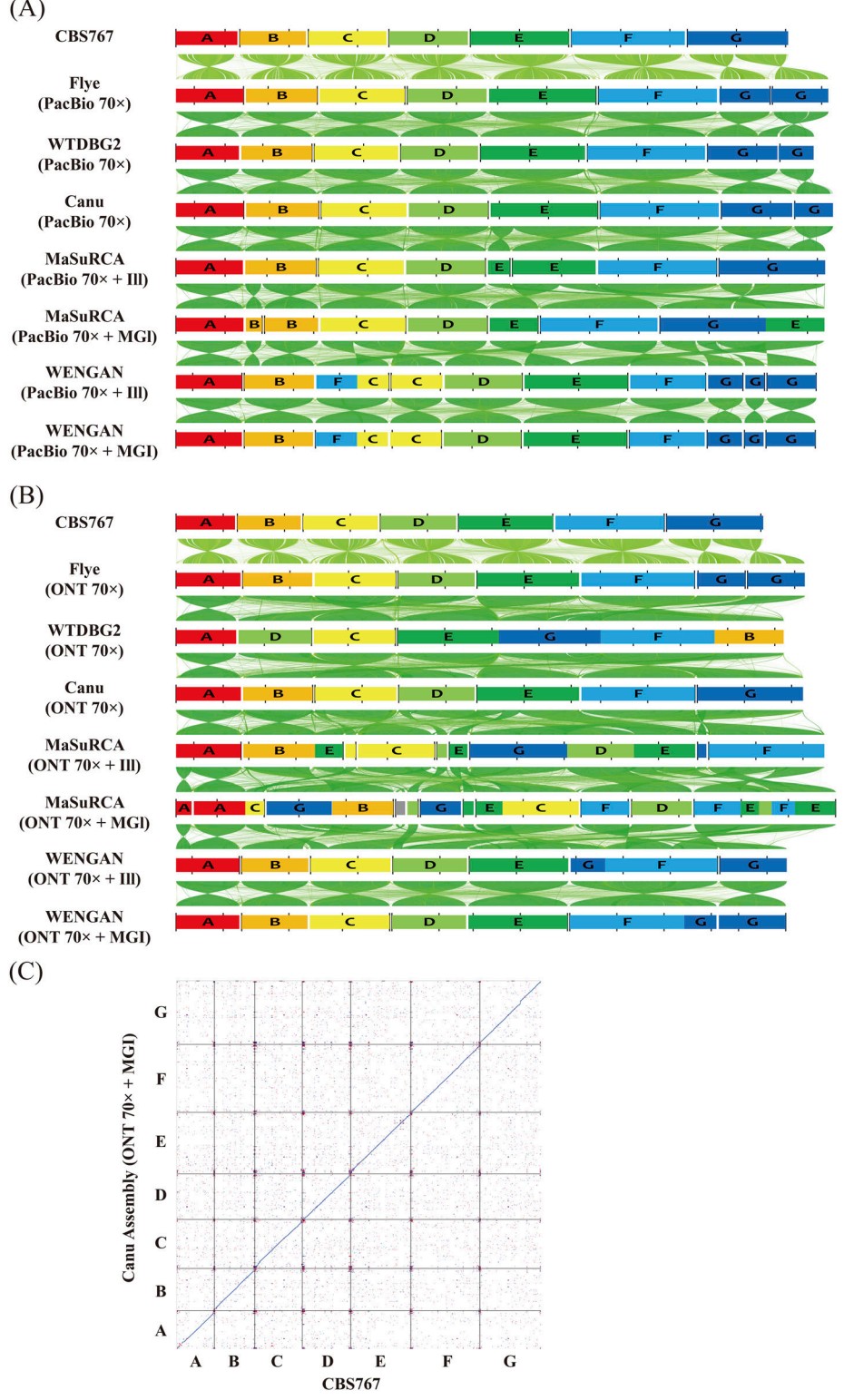

**Figure 3. Chromosomal structures of assembly results.**
The results of each TGS-based assembly are presented in the form of multiple sequence alignments and comparisons using AliTV and LAST. The assembly structures are marked by a chromosome name and color that matches the part of the reference genome. **(A, B)** Each contig of assembly is distinguished by the block separated by white space in (A) PacBio 70× assembly and (B) ONT 70× assembly. **(C)** Sequence alignment between the best-performing ONT 70× Canu assembly and the reference genome, *D. hansenii* CBS767, consisting of seven chromosomes (A, B, C, D, E, F, and G).

annotation, even without the assistance of secondary sequencing platforms (such as Illumina or MGI). However, the fragmentation in long repetitive regions observed in both assembly programs suggests the necessity of longer TGS reads such as ONT, which can cover the long range of those complex regions (e.g., structural variants). The ONT-based assembly results from non-hybrid assemblers showed more highly continuous contigs than PacBio-based assemblies and produced the best-fit assembly (Canu 70×

**Table 1. Assembly results from (A) PacBio and (B) Nanopore at 70× coverage.**

**(A)**

| PacBio 70× | Total length (bp) | Tig num. | N50 (bp) | BUSCO completeness | Mercury completeness |
|---|---|---|---|---|---|
| Flye (+ Ill) | 12924438 | 8 | 1612849 | C:90.0%[S:90.0%, D:0.0%], F:0.1%, M:9.9%, n:758 | 99.76% |
| Flye (+ MGI) | 12922222 | 8 | 1612842 | C:90.0%[S:90.0%, D:0.0%], F:0.1%, M:9.9%, n:758 | 99.74% |
| WTDBG2 (+ Ill) | 12633374 | 8 | 1574041 | C:89.8%[S:89.8%, D:0.0%], F:0.1%, M:10.1%, n:758 | 99.32% |
| WTDBG2 (+ MGI) | 12629300 | 8 | 1574044 | C:90.0%[S:90.0%, D:0.0%], F:0.1%, M:9.9%, n:758 | 99.32% |
| Canu (+ Ill) | 13015402 | 8 | 1612759 | C:90.1%[S:90.1%, D:0.0%], F:0.1%, M:9.8%, n:758 | 99.78% |
| Canu (+ MGI) | 13015282 | 8 | 1612729 | C:90.1%[S:90.1%, D:0.0%], F:0.1%, M:9.8%, n:758 | 99.75% |
| MaSuRCA (+ Ill) | 12853850 | 8 | 1692499 | C:90.1%[S:90.0%, D:0.1%], F:0.1%, M:9.8%, n:758 | 99.76% |
| MaSuRCA (+ MGI) | 12838108 | 8 | 1730471 | C:89.8%[S:89.7%, D:0.1%], F:0.1%, M:10.1%, n:758 | 99.44% |
| WENGAN (+ Ill) | 12576158 | 10 | 1472709 | C:90.2%[S:90.1%, D:0.1%], F:0.3%, M:9.5%, n:758 | 99.35% |
| WENGAN (+ MGI) | 12555145 | 10 | 1472717 | C:89.9%[S:89.8%, D:0.1%], F:0.3%, M:9.8%, n:758 | 99.29% |

**(B)**

| Nanopore 70× | Total length (bp) | Tig num. | N50 (bp) | BUSCO completeness | Mercury completeness |
|---|---|---|---|---|---|
| Flye (+ Ill) | 13005287 | 8 | 1626946 | C:89.4%[S:89.4%, D:0.0%], F:0.4%, M:10.2%, n:758 | 99.67% |
| Flye (+ MGI) | 13002089 | 8 | 1626972 | C:90.2%[S:90.2%, D:0.0%], F:0.1%, M:9.7%, n:758 | 99.69% |
| WTDBG2 (+ Ill) | 12782527 | 4 | 8218140 | C:88.8%[S:88.8%, D:0.0%], F:0.4%, M:10.8%, n:758 | 99.37% |
| WTDBG2 (+ MGI) | 12781418 | 4 | 8217105 | C:89.8%[S:89.8%, D:0.0%], F:0.1%, M:10.1%, n:758 | 99.41% |
| Canu (+ Ill) | 13040800 | 7 | 2175762 | C:88.9%[S:88.9%, D:0.0%], F:0.8%, M:10.3%, n:758 | 99.69% |
| Canu (+ MGI) | 13040511 | 7 | 2175685 | C:90.4%[S:90.4%, D:0.0%], F:0.1%, M:9.5%, n:758 | 99.72% |
| MaSuRCA (+ Ill) | 13348856 | 9 | 2452519 | C:89.8%[S:86.9%, D:2.9%], F:0.1%, M:10.1%, n:758 | 99.77% |
| MaSuRCA (+ MGI) | 13487146 | 11 | 2176961 | C:89.8%[S:85.4%, D:4.4%], F:0.3%, M:9.9%, n:758 | 99.18% |
| WENGAN (+ Ill) | 12650894 | 7 | 1692220 | C:90.1%[S:90.1%, D:0.0%], F:0.1%, M:9.8%, n:758 | 99.49% |
| WENGAN (+ MGI) | 12629331 | 7 | 1690156 | C:89.7%[S:89.7%, D:0.0%], F:0.1%, M:10.2%, n:758 | 99.43% |

assembly). These results may, however, be biased regarding two different aspects: chromosomal structure and completeness. For example, WTDBG2 dramatically decreases the amount of calculation in the assembly process by compressing homopolymers in a representative single base with a slight loss of sensitivity and specificity (Miller et al, 2008; Au et al, 2012; Ruan & Li, 2020). Our results, however, demonstrate the significantly low accuracy of WTDBG2 assemblies from ONT sequences, which are relatively vulnerable to homopolymer bias. Furthermore, in ONT 70× dataset, contigs of WTDBG2 revealed arbitrary chromosomal fusion merged in the telomeric region, where four aligned reference chromosomes were integrated into one contig. Thus, those deteriorative assembly results alert warning that certain features of sequencing technology can interfere with optimal assembly depending on the complexity reduction method, suggestive of the negative association between sequencing property and assembly process.

With regard to hybrid assemblers, MaSuRCA produced a higher ratio of chimeric contigs than WENGAN, particularly in ONT-based assemblies. Because both assembly tools employ SGS-based pre-assembly processes, more chimeric contigs were found in hybrid assemblies than in non-hybrid assemblies, even at the highest TGS

coverage (70×) in this experiment. The reference-close chromosomal structures were only seen in the 50× PacBio + Illumina read-based MaSuRCA assembly, which scored low on BUSCO completeness (78.4%). In addition, the ONT-based MaSuRCA assemblies, which adopted partial alignment-based greedy extension, indicate that the low accuracy of template TGS reads and the relatively short super-reads could render the assembly results more chimeric and incapable of repeat resolving (Bresler et al, 2013), in contrast to the WENGAN assembler that incorporates a pseudo–paired-end-alignment-based repeat sequence solution (Di Genova et al, 2021). The ONT-based Canu assembly, which shares the CABOG consensus algorithm with MaSuRCA, but strictly controls the formation of chimeric contigs, indicates the limitations of MaSuRCA's aggressive consensus strategy followed by the short-read–based pre-assembly method (Miller et al, 2008; Koren et al, 2017; Zimin et al, 2017). In that respect, the synergy of low-accurate TGS reads and the MaSuRCA assembler can drive highly biased assembly results generated from a lack of a clear rationale for sequence overlap. It is therefore important to select the appropriate pipeline according to the respective data properties.

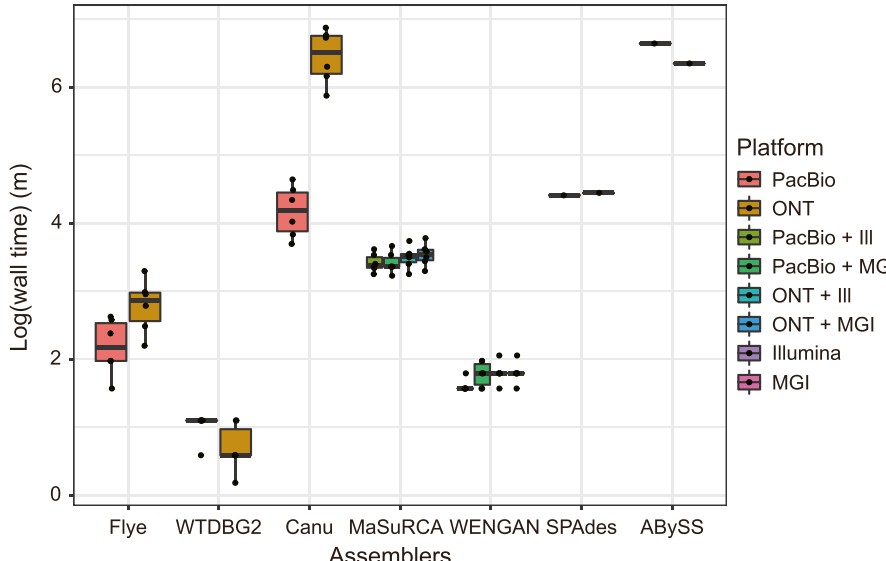

**Figure 4. Time consumption of each assembly process.**
Each assembly process was assessed for real wall time by the logarithmic value of minutes (m). Variations according to the six different coverages were classified according to the type of raw dataset, indicated by different colors. For the SPAdes and ABySS, which proceeded through the short-read–only assembly pipelines, their time consumption is displayed as a single dot according to the SGS platform (Illumina or MGI).

It is also meaningful to pay attention to the results of short-read–only assemblies to understand the difference in roles between Illumina and MGI. The assemblies of SPAdes and ABySS with Illumina reads perform better in metrics of completeness (BUSCO value) based on the higher contiguity (contig number and N50). Interestingly, this platform-specific property of SGS technology is observed as opposite in comparison with the quality metrics of non-hybrid assemblers; the polishing process with the MGI read always generates equal or higher BUSCO completeness than that of Illumina. We found that our results were in line with a previous report on the platform-specific variants between Illumina NovaSeq 6000 and MGI DNBSEQ-T7 as whole-genome sequencing platforms; Illumina NovaSeq 6000 exhibited higher indel frequency for a length of 1 bp, but MGI DNBSEQ-T7 showed higher indel frequencies for indels above 8 bp (Jeon et al, 2021). In other words, the Illumina

data are advantageous to extend contig but are more vulnerable to single-base indel errors than the MGI data. Although MaSuRCA assemblies show inconsistent quality in both SGS datasets, assemblies of WENGAN also represent equal or higher completeness in Illumina-based results. Thus, the comparison between the two SGS platforms suggests that MGI reads are insufficient as sole assembly datasets, but with template TGS-based assembly, they are adequate for secondary sequencing in the polishing process from the perspective of accuracy and cost-efficiency.

The best assembly result was the ONT 70×-based MGI-polished Canu assembly, which not only scored high on chromosomal structure, contiguity metrics, and completeness, but also consumed more time than all other assemblers. A scientist pressed for time who wants to complete a genome analysis rather than prioritize accurate chromosomal structure might therefore select an assembly pipeline

**Table 2. Classification of assembly strategies.**

| Assembler | Error correction | Contig extension | Complexity reduction |
|---|---|---|---|
| Flye | | Graph-based disjointing correction | Disjointig construction |
| WTDBG2 | Direct | Fuzzy Bruijn graph | Hash table based on the *k*-mer block (bin) |
| | | Partial order alignment | Non-redundant *k*-mer removing |
| Canu | Hierarchical | Best overlap graph | *tf-idf* weighted MHAP |
| | | CABOG | |
| | | PBDAG-CON | |
| MaSuRCA | Hybrid | de Bruijn | Super-read construction |
| | | CABOG | |
| WENGAN | | de Bruijn | Synthetic scaffolding graph |
| | | Partial order alignment | |
| SPAdes | Short-read only | Multisized de Bruijn | — |
| ABySS | | de Bruijn | — |
| | | Paired-end–based contig extension | |

that uses Flye or WENGAN. However, if complete results are required, even if it takes a long time, Canu, which reduces the occurrence of chimeric contigs, might be the better choice.

One of the unexpected results of our study is that gradually increasing coverage of TGS read depth was not in proportion with assembly quality. This counterintuitive phenomenon was observed in both hybrid and non-hybrid assemblers: ONT-based Canu (40× versus 50×) and PacBio-based MaSuRCA (Illumina 50× versus 60× and MGI 40× versus 50×). As those sudden drops in assembly quality occurred inconsistently across datasets and assembly programs, we can infer the existence of adequate coverage for each assembly program, which can provide reasonable assembly results without effects on the platform-specific variants. Our observation thus suggests that an excessive level of coverage could have rather negative effects on the overall quality of assembly.

In summary, our study revealed the effect of the characteristics of sequencing technology and sequencing depth on assembly algorithm by analyzing the results of the combination of various dataset conditions and assembly programs in the whole-genome assembly process of the repetitive yeast genome. The assembly patterns also suggest that pipelines consisting of non-cooperative sequencing techniques and assembly programs lead to the whole-genome reconstruction of low quality. Overall, this represents the possibility of presenting an efficient assembly strategy according to the user's purpose even for the de novo yeast whole-genome assembly.

# Materials and Methods

### Yeast strain and culture conditions

The strain *D. hansenii* KCTC27743 was obtained from the Korean Collection for Type Cultures (KCTC; https://kctc.kribb.re.kr). This strain was originally isolated from the Korean traditional fermented product "nuruk." This strain was cultivated in YPD (1% yeast extract, 2% Bacto-peptone, and 2% glucose) medium.

### Sequencing and library preparation for PacBio, ONT, and Illumina data

High-quality chromosomal DNA was obtained from yeast spheroplast, generated by treatment with 0.5 KU of zymolyase (100T; MP Bio-medicals) using the spooling method. PacBio long-read sequencing data for *D. hansenii* KCTC27743 were generated by the Sequel (Pacific Biosciences) platform. To construct libraries for sequencing, more than 10 μg of high-quality/high molecular weight genomic DNA was sheared up to 20 kbp using a 26-g blunt needle (SAI Infusion Technologies). Sheared genomic DNA was treated for damage repair and end-repair, adapter ligation, and size selection with a BluePippin system (Sage Science), generating SMRTbell template libraries.

ONT sequencing libraries were prepared according to the SQK-LSK109-MinION gDNA Sequencing Kit (Oxford Nanopore Technologies) protocol. The 5 μg of libraries was added to the R7.3 flow cells of the ONT MinION for a 24-h run and constructed from only 1D data. Paired-end 150-bp Illumina data for *D. hansenii* KCTC27743 were generated using an Illumina NovaSeq 6000 platform (Illumina). Libraries with genomic DNAs were prepared using the TruSeq Nano

DNA Sample Prep Kit (Illumina) according to the manufacturer's protocols. Sheared DNA fragments were further processed by end-repairing, A-tailing, adapter ligation, and amplification with clean-up.

### Sequencing and library preparation for MGI data

Chromosomal DNA was extracted from the cell sample using the QIAGEN Genomic-tip 20/G (QIAGEN) following the manufacturer's instructions. For the MGI sequencing, the libraries were constructed to be suitable for PE150 according to the MGI FS DNA library prep set (MGI). The amplified libraries were sequenced using DNBSEQ-T7 (MGI) with a PE read length of 150 bp.

### Short-read trimming and genome size estimation

Raw datasets from the short-read platforms (Illumina and MGI) were trimmed by Trim Galore! (ver. 0.6.7) (Babraham Bioinformatics) using the parameter "--quality 30 --max_n 0 --length 120" to remove reads with low quality. Histograms obtained with five units of $k$-mer ($k$ = 15, 17, 19, 21, and 23) with Jellyfish (ver. 2.3.0) (Marçais & Kingsford, 2011) were used for the subsequent estimation of genome size through GenomeScope (http://qb.cshl.edu/genomescope) (Vurture et al, 2017). Based on the tool's recommendation, the estimated genome size (12.53 Mbp) that produced the highest model fit (99.32%) with the $k$-mer ($k$ = 15) of the Illumina dataset was finally selected for the further genome assembly process (Ranallo-Benavidez et al, 2020).

### Long-read subsampling and quality check

The Rasusa program (ver. 0.3.0) (Hall, 2022) can consider both read count and total read length. This characteristic of the tool made low-biased random subsamples of each long-read sequence (with PacBio Sequel and ONT MinION) at 20×, 30×, 40×, 50×, 60×, and 70×. The read number, total base number, and N50 of each coverage group were further estimated with the LongQC (ver. 1.2.0b) (Fukasawa et al, 2020) and NanoPlot (ver. 1.39.0) (De Coster et al, 2018). The homomer counting was conducted using our custom Python script "homo-mer_count.py" (https://github.com/MSjeon27/homomer_count).

### Genome assembly and correction

All assembly processes were performed on a 64-bit ×86 HP DL380 Gen10 machine with 32 cores at 3.2 GHz and 64 GB of RAM, running Linux kernel (ver. 3.10.0). The subsampled long reads (PacBio Sequel and ONT MinION) and short reads (Illumina NovaSeq 6000 and MGI DNBSEQ-T7) were subsequently assembled by seven assembly programs: the Flye (ver. 2.8.1-b1676), WTDBG2 (ver. 2.5), Canu (ver. 2.1), MaSuRCA (ver. 4.0.5), WENGAN (ver. 0.2), SPAdes (ver. 3.14.0), and ABySS (ver. 2.3.5). The assemblers except for the SPAdes and ABySS processes estimated genome size (12.53 Mbp) as their essential argument with 32 threads option. The Canu operates without a set of threads because it automatically allocates maximal available computational resources for each stage (Koren et al, 2017). Each assembly process was checked for CPU and memory usage at the 1-min interval with psrecord (ver. 1.2; https://github.com/astrofrog/psrecord). Two types of SGS were subsampled in 4 million reads using seqtk (ver. 1.3-r106; https://github.com/lh3/seqtk). To

increase the accuracy of contigs, one of the SGS reads (Illumina NovaSeq 6000 and MGI DNBSEQ-T7) was first aligned to a draft assembly with Bowtie2 (ver. 2.3.5), using the "--very-sensitive –all –no-unal" option (Langmead & Salzberg, 2012). The produced SAM files were then converted into sorted BAM files with SAMtools (ver. 1.10-37-g2e4d43a) (Danecek et al, 2021). Post-assembly polishing was processed in two cycles using PILON (ver. 1.23) (Walker et al, 2014) for each draft assembly. After following these workflows, only the contigs up to a length cutoff of 100,000 bp were selected from each assembly output to exclude potential misassemblies.

### Genome assembly assessment of contiguity and completeness

The statistical assessment of polished assemblies was conducted with Quast (ver. 5.0.2) using the default option (Mikheenko et al, 2018). Quast reports significant metrics such as total genome size, N50, N75, GC content, maximum contig length, and reference-based parameters for each assembly result. The most representative strain of the yeast *D. hansenii* CBS767 (accession number, GCA_000006445.2) was used as the reference genome. LAST and AliTV furtherly visualized the chromosomal structure conformation and the reference-based structural variants such as amplification, inversion, or translocation. LAST's alignment generator lastal (ver. 1266) generated the alignment using the default parameters and the "-cR01" option (Kiełbasa et al, 2011). The lastal output was parsed to last-dotplot, to generate full-genome visualization between the pairs of different yeast species. The multiple alignments among assembly results and the reference genome were also pre-processed in JavaScript object notation (JSON) format using AliTV (ver. 1.0.6) and visualized on https://alitvteam.github.io/AliTV/d3/AliTV.html (Ankenbrand et al, 2017).

The quantitative evaluations of genome completeness were processed using Benchmarking Universal Single-Copy Orthologs (BUSCO, ver. 3.1.0), with the Fungi database (fungi_odb9) and the default parameters (Seppey et al, 2019). To verify the effects of the PILON processes, we compared the before- and after-polishing assembly contigs, thereby identifying the ratios of absent, single, double, or fragmented genes. In addition, the completeness of the draft assemblies without the reference genome was measured by aligning trimmed Illumina data to reconstructed genomes using Merqury (ver. 1.3) (Rhie et al, 2020).

## Data Availability

The raw sequencing data of the *Debaryomyces hansenii* KCTC27743 are available from SRA (SRR19837491–SRR19837494). The assembly file of *D. hansenii* KCTC27743 is available from GenBank accession (CP045111–CP045117), Project number (PRJNA576775), and Sample number (SAMN13008417). The source code of the custom Python script homomer_count.py is available on GitHub (https://github.com/MSjeon27/homomer_count) with an MIT License.

## Supplementary Information

## Acknowledgements

This work was supported by the National Research Foundation of Korea (2022R1A2C4002058), the Cooperative Research Program for Agriculture Science & Technology Development (PJ01710102) funded by the Rural Development Administration, and Korea Institute of Marine Science and Technology Promotion (20220532) funded by the Ministry of Oceans and Fisheries.

### Author Contributions

M-S Jeon: conceptualization, data curation, formal analysis, validation, investigation, visualization, methodology, project administration, and writing—original draft, review, and editing.
DM Jeong: resources, data curation, formal analysis, investigation, methodology, and writing—original draft, review, and editing.
H Doh: data curation, formal analysis, investigation, and writing—original draft, review, and editing.
HA Kang: formal analysis, supervision, investigation, project administration, and writing—original draft, review, and editing.
H Jung: conceptualization, data curation, formal analysis, supervision, investigation, methodology, project administration, and writing—original draft, review, and editing.
S Eyun: conceptualization, resources, data curation, formal analysis, supervision, funding acquisition, investigation, methodology, project administration, and writing—original draft, review, and editing.

### Conflict of Interest Statement

The authors declare that they have no conflict of interest.

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
