## [Reviewer comments · Life Science Alliance]

Life Science Alliance

A practical comparison of the next-generation sequencing platform and assemblers using yeast genome

Min-Seung Jeon, Da Min Jeong, Huijeong Doh, Hyun Ah Kang, Hyungtaek Jung and Seong-il Eyun

DOI: <https://doi.org/10.26508/lsa.202201744>

Corresponding author(s): Prof. Seong-il Eyun (Chung-Ang University); Hyungtaek Jung

Review Timeline:

Submission Date:	2022-09-28
Editorial Decision:	2022-10-31
Revision Received:	2022-12-27
Editorial Decision:	2023-01-19
Revision Received:	2023-01-25
Accepted:	2023-01-25

Transaction Report:

October 31, 2022

Re: Life Science Alliance manuscript #LSA-2022-01744-T

Seong-il Eyun
Department of Life Science, Chung-Ang University

Dear Dr. Eyun,

Thank you for submitting your manuscript entitled "A practical comparison of the next-generation sequencing platform, depth, and assembly software using yeast genome" to Life Science Alliance. The manuscript was assessed by expert reviewers, whose comments are appended to this letter. We invite you to submit a revised manuscript addressing the Reviewer comments.

Thank you for this interesting contribution to Life Science Alliance. We are looking forward to receiving your revised manuscript.

Sincerely,

B. MANUSCRIPT ORGANIZATION AND FORMATTING:

Reviewer #1 (Comments to the Authors (Required)):

Dear

The manuscript entitled 'A practical comparison of the next-generation sequencing platform, depth and assembly software using yeast genome' by Jeon et al. describes the benchmarking of different combinations of sequencing platforms and assembly tools to generate yeast genome assembly. The methods and results are clearly written and easy to understand. The results will be of use to relevant researchers in the field. I have only two minor comments.

Comment.

1. In Figure 4, the authors suggest that a significant difference in the length of the raw TGS data length could explain the effects on the use of time. In this regard, I think that the longer the raw data, the shorter the assembly time, but the authors interpret the results in reverse and insist that the longer the raw data, the more time is consumed. Could the authors add some references or evidence to support their claim?
2. Line 340, what do the authors mean by 'negative synergy between sequencing and assembly'? I find the phrase difficult to understand.

Reviewer #2 (Comments to the Authors (Required)):

The reported work compared the accuracy, efficiency and time consumption for de novo assembling genomes using sequencing reads generated by the second generation sequencing technology, third generation sequencing technology or second generation sequencing plus third generation sequencing technologies, using yeast genome as an example and with different sequencing coverages. The results are informative, but the data should be better presented.

Major comments:

1. In lines 122-125, the authors claimed that the objective of this work is to provide optimal sequencing standards for de novo yeast genome sequencing, what is special for the yeast genome? Could the same standards also provide guidance for the assembly of genomes from other species?
2. There are many tools for short-read assembly, other commonly used ones should be included for comparison.
3. In Figure 2B, why there is a sudden drop in BUSCO scores of PacBio + Ill and PacBio + MGI?
4. Figure 2D provided too little information.
5. Better to show time in minutes in Figure 4 to avoid negative log values.

Minor points:

1. Figures and Tables should not be placed before the corresponding result sessions.
2. The labels in Supplementary Figure 2 are too small.
3. "Polishing" should be "Polished" in Figure 2A.

Reviewer #1 (Comments to the Authors (Required)):

Dear

The manuscript entitled 'A practical comparison of the next-generation sequencing platform, depth and assembly software using yeast genome' by Jeon et al. describes the benchmarking of different combinations of sequencing platforms and assembly tools to generate yeast genome assembly. The methods and results are clearly written and easy to understand. The results will be of use to relevant researchers in the field. I have only two minor comments.

Comment.

1. In Figure 4, the authors suggest that a significant difference in the length of the raw TGS data length could explain the effects on the use of time. In this regard, I think that the longer the raw data, the shorter the assembly time, but the authors interpret the results in reverse and insist that the longer the raw data, the more time is consumed. Could the authors add some references or evidence to support their claim?

In response to this comment, in this revision we examined the time-consuming per stage in three non-hybrid assemblers (Figure S5 and Table S6) (Line 403). Once measuring the time required for each stage of the two platforms, it was commonly found in all three assemblies that the time differences were more pronounced in the contig extension stage than in the correction stage which is related to the accuracy of the read. However, we found that the differences between the two TGS platforms were in both accuracy and length, so we cannot specify which one is the determinant. Therefore, the longer the raw data length, the more time it is likely to take, but the results described that the time differences were more pronounced in the contig extension stage than in the correction stage.

2. Line 340, what do the authors mean by 'negative synergy between sequencing and assembly'? I find the phrase difficult to understand.

Following the reviewer's suggestion, we remove the inexplicit phrase. The contig extension stage of WTDBG2 is based on de Bruijn graph which is composed with homopolymer compressed sequence information. However, nanopore sequencing platform such as R7 nanopore is vulnerable to the homopolymer error. Thus, the combination of sequencing reads with homopolymer error and assembly algorithm of WTDBG2 made the deteriorative assembly results in the prospect of the accuracy, even creating chimeric scaffolds. We revised and added the sentence on Line 498 (Page 21):

"Thus, those deteriorative assembly result alert warning that certain features of sequencing technology can interfere with optimal assembly depending on the complexity

reduction method, suggestive of the negative association between sequencing property and assembly process.”

Reviewer #2 (Comments to the Authors (Required)):

The reported work compared the accuracy, efficiency and time consumption for de novo assembling genomes using sequencing reads generated by the second generation sequencing technology, third generation sequencing technology or second generation sequencing plus third generation sequencing technologies, using yeast genome as an example and with different sequencing coverages. The results are informative, but the data should be better presented.

Major comments:

1. In lines 122-125, the authors claimed that the objective of this work is to provide optimal sequencing standards for de novo yeast genome sequencing, what is special for the yeast genome? Could the same standards also provide guidance for the assembly of genomes from other species?

At the beginning of this projects, our study goal finds the optimal assembly pipeline for yeast genome. We sequenced more than 50 yeast genomes and do works the assembly pipeline. However, as mentioned in Magoc et al (2013) in Line 460, the optimal assembly can vary according to the properties of biological sequences, even in low-complex bacterial genomes and it can therefore be difficult to adjust the system to the proper genomic reconstruction, depending on which species is analyzed. We faced that the major difficult point of yeast assembly process is the variable telomeric sequences. These various sequences of telomere regions make it difficult to distinguish whether the repetitive sequence occurs inside the genome (tandem repeats) or at the end (telomeric sequence) based on the database of the repeat elements (cf. RepBase). We thus not only checked the traditional metrics for assembly quality (the assessment by QUAST and BUSCO) but also compared the assembled chromosomal structure vizualized by AliTV.

2. There are many tools for short-read assembly, other commonly used ones should be included for comparison.

Following the reviewer's suggestion, we added one more short-read-based assembler and included the results in this revised version. We added ABySS because we have focused on the comparison of assemblers using TGS with the high quality. In the results of the ABySS assembler using only the short-read sequence, Illumina-based assembly showed better indicators in both continuity and completeness than that of MGI. Therefore, this is further described in Discussion as more definite result of the difference in roles between Illumina and MGI.

3. In Figure 2B, why there is a sudden drop in BUSCO scores of PacBio + Ill and PacBio + MGI?

We observed a sudden drop in not only BUSCO scores but also merqury completeness of both PacBio + Illumina and PacBio + MGI. Considering that Merqury completeness is an indicator of how well raw data in short-read sequencing is aligned to assembly, we could suspect that the assembly was partially poorly constructed due to severe fragmentation in certain parts of MaSuRCA's short-read-based assembly. The consecutive synteny analysis revealed that this 'sudden dropped assemblies' were lacking one of the chromosomes (chromosome number 4) compared with the well-assembled Canu 70X Nanopore and MGI-based assembly (Figure S3). Therefore, we supposed that the contig fragmentation caused by misassembly has become so severe that the quality has been lowered. We revised and added the sentence on Line 292 (Page 11):

“The further study of synteny analysis represented this sudden drop of BUSCO value was caused by the severe fragmentation of scaffold in assembly procedure (Figure S3).”

4. Figure 2D provided too little information.

We removed Figure 2D and revised Figure S3. The main purpose of Fig. 2D showed that MaSuRCA assembly process is significantly duplicated or fragmented due to short-read-based greedy extension. In this revised version, Figure S3 describes the fragmentation and misassembly of the MaSuRCA assembly.

5. Better to show time in minutes in Figure 4 to avoid negative log values.

We revised the time consuming to the logarithmic minute ($\log(M)$).

Minor points:

1. Figures and Tables should not be placed before the corresponding result sessions.

We revised all figures and tables after the corresponding result sessions.

2. The labels in Supplementary Figure 2 are too small.

In this revised version, the font size in all figures is increased.

3."Polishing" should be "Polished" in Figure 2A.

We revised the text, "Polishing" to "Polished" in Figure 2A (Line 306)

January 19, 2023

RE: Life Science Alliance Manuscript #LSA-2022-01744-TR

Prof. Seong-il Eyun
Chung-Ang University
Department of Life Science
84, Heukseok-ro, Dongjak-gu, Seoul, Republic of Korea
Seoul 06974
Korea, Republic of (South Korea)

Dear Dr. Eyun,

Thank you for submitting your revised manuscript entitled "A practical comparison of the next-generation sequencing platform and assemblers using yeast genome". We would be happy to publish your paper in Life Science Alliance pending final revisions necessary to meet our formatting guidelines.

- please upload all figure files as individual ones, including the supplementary figure files; all figure legends should only appear in the main manuscript file. Please remove your figures from the manuscript text
- please add the Twitter handle of your host institute/organization as well as your own or/and one of the authors in our system
- please add an Author Contributions section to your main manuscript text
- please add your main, supplementary figure, and table legends to the main manuscript text after the references section
- please upload a clean version of your paper without the track changes
- please add callouts for Figures 2C, 3A-C and S1B to your main manuscript text

A. FINAL FILES:

B. MANUSCRIPT ORGANIZATION AND FORMATTING:

Sincerely,

Reviewer #2 (Comments to the Authors (Required)):

The authors have addressed all my questions, now the manuscript has been improved and could be accepted for publication.

January 25, 2023

RE: Life Science Alliance Manuscript #LSA-2022-01744-TRR

Prof. Seong-il Eyun
Chung-Ang University
Department of Life Science
84 HeukSeok-ro
Dongjak-gu
Seoul 06974
Korea, Republic of (South Korea)

Dear Dr. Eyun,

Thank you for submitting your Resource entitled "A practical comparison of the next-generation sequencing platform and assemblers using yeast genome". It is a pleasure to let you know that your manuscript is now accepted for publication in Life Science Alliance. Congratulations on this interesting work.

DISTRIBUTION OF MATERIALS:

Again, congratulations on a very nice paper. I hope you found the review process to be constructive and are pleased with how the manuscript was handled editorially. We look forward to future exciting submissions from your lab.

Sincerely,
